# The Glycoprotein M6a Is Associated with Invasiveness and Radioresistance of Glioblastoma Stem Cells

**DOI:** 10.3390/cells11142128

**Published:** 2022-07-06

**Authors:** Marie Geraldine Lacore, Caroline Delmas, Yvan Nicaise, Aline Kowalski-Chauvel, Elizabeth Cohen-Jonathan-Moyal, Catherine Seva

**Affiliations:** 1INSERM UMR 1037 Cancer Research Center of Toulouse (CRCT), University Paul Sabatier Toulouse III, Avenue Hubert Curien, 31100 Toulouse, France; mglacore@gmail.com (M.G.L.); delmas.caroline@iuct-oncopole.fr (C.D.); yvan.nicaise@inserm.fr (Y.N.); a.chauvel@free.fr (A.K.-C.); moyal.elizabeth@iuct-oncopole.fr (E.C.-J.-M.); 2IUCT-Oncopole, Avenue Hubert Curien, 31100 Toulouse, France

**Keywords:** glioblastomas, invasion, radioresistance, cancer stem cells, GPM6A, PTPRZ1

## Abstract

Systematic recurrence of glioblastoma (GB) despite surgery and chemo-radiotherapy is due to GB stem cells (GBSC), which are particularly invasive and radioresistant. Therefore, there is a need to identify new factors that might be targeted to decrease GBSC invasive capabilities as well as radioresistance. Patient-derived GBSC were used in this study to demonstrate a higher expression of the glycoprotein M6a (GPM6A) in invasive GBSC compared to non-invasive cells. In 3D invasion assays performed on primary neurospheres of GBSC, we showed that blocking GPM6A expression by siRNA significantly reduced cell invasion. We also demonstrated a high correlation of GPM6A with the oncogenic protein tyrosine phosphatase, PTPRZ1, which regulates GPM6A expression and cell invasion. The results of our study also show that GPM6A and PTPRZ1 are crucial for GBSC sphere formation. Finally, we demonstrated that targeting GPM6A or PTPRZ1 in GBSC increases the radiosensitivity of GBSC. Our results suggest that blocking GPM6A or PTPRZ1 could represent an interesting approach in the treatment of glioblastoma since it would simultaneously target proliferation, invasion, and radioresistance.

## 1. Introduction

Glioblastoma (GB) is an aggressive and infiltrating tumor of the central nervous system (CNS) with a median overall survival of less than two years [1]. The invasive phenotype makes surgical resection difficult and incomplete. The recurrence of glioblastoma is systematic despite surgery and chemo-radiotherapy. Certain clinical characteristics, including the localization and difficult surgical access, can favor this high recurrence. In particular, tumors in contact with the periventricular zone are more aggressive and have a decreased overall survival (OS) rate when compared to cortical tumors [2,3]. Patients with multiple lesion glioblastoma also have a poor prognosis and present a shorter progression-free survival and OS [4]. Based on genetic characteristics, GB have been classified into different subtypes including the classical, proneural, and mesenchymal subtype, the latter being the most aggressive and resistant to radio-chemotherapy. GB stem cells (GBSC), which have the capacity for self-renewal and contribute to tumor initiation, are particularly resistant to therapies and have also been involved in recurrence [5,6]. GBSC are characterized by a high invasive potential, but the mechanisms that regulate their invasive capacity are not fully understood. Currently, there is no clinical therapy that targets GBSC. Therefore, deciphering the molecular mechanisms responsible for the resistance and invasiveness of GBSC is critically needed to develop new effective therapies for GB.

In the present paper, we focused on the glycoprotein M6a (GPM6A), a four transmembrane protein that belongs to the proteolipid protein (PLP) family. GPM6A is highly expressed in the CNS, and transcriptomic datasets per cell type, publicly available in the Human Protein Atlas Database, have shown that GPM6A is particularly well-expressed in astrocytes, oligodendrocyte precursor cells, and microglia. In normal neuronal cells, GPM6A accumulates in lipid raft domains and acts as a transducer for extracellular signals such as laminin [7]. It plays an important role in neurite outgrowth, filopodia formation, and neuronal migration. GPM6A overexpression in neuronal and non-neuronal cells induces extensive formation of filopodia-like protrusions, presumably through the activation of the small GTPase Rac1 and kinases such as PAK1, Src, and MAPK. GPM6A has also been shown to be involved in the proliferation of neuronal stem cells and non-neuronal cells [7,8,9,10,11]. The role of GPM6A in cancer cells has not been extensively studied. In lymphoid leukemias, GPM6A and GPM6B are overexpressed and act as oncogenes in the development of these malignancies [12]. In sporadic non-functioning pituitary adenomas, Falch et al. reported a higher expression of GPM6A in fast-growing compared to slow-growing adenomas [13]. In colorectal cancer, the up-regulation of GPM6A was closely related to a poorer overall survival. In addition, a higher expression of GPM6A was observed in the poorly differentiated compared to the highly differentiated colorectal carcinoma tissues [14,15].

In the Human Protein Atlas Database, the RNA expression overview from The Cancer Genome Atlas shows an enrichment of GPM6A and GPM6B RNA in glioma. However, the role of these proteins in glioblastoma has never been studied. Only one publication has reported that the high expression of GPM6B in patient samples allowed discrimination between glioblastoma and meningioma cases [16].

In this study, we report that GPM6A is overexpressed in highly invasive GBSC. We also demonstrate that targeting GPM6A represses cell invasion, decreases neurosphere formation, and increases the radiosensitivity of GBSC.

## 2. Materials and Methods

### 2.1. GB Patient-Derived Cells

GB biopsies were performed in the Neurosurgery Department at Toulouse University Hospital under an approved clinical protocol (ethical code 12TETE01, ID-RCB number 2012-A00585-38, date of approval: 7 May 2012). Written informed consents were obtained for all the patients. WHO was used to classify the tumors as GB. The GBSC were isolated from GB specimens and cultured as described by Avril et al. [17] in DMEM-F12 (GIBCO, Waltham, MA, USA) including N2 and B27 (LifeTechnologies, Carlsbad, CA, USA), and EGF and FGF2 (Peprotech, East Windsor, NJ, USA) at 37 °C in a CO_2_ incubator (5%). The GBSC used in the study (3 mesenchymal: GSC08, GSC10, and GSC14 and 11 proneural: GSC01, GSC02, GSC03, GSC04, GSC05, GSC06, GSC07, GSC09, GSC11, GSC12, and GSC13) have been characterized by the overexpression of stem cell markers (SOX2, OLIG2), their ability to differentiate into neural lineages, self-renewal, and their tumorigenic potential in vivo (Appendix A). Neurospheres are cultured for fewer than 12 passages to keep stem characteristics.

### 2.2. Three-Dimensional Invasion Assays

Three-dimensional invasion assays have been previously described by Vinci M. et al. [18]. Briefly, the cells were seeded into ultra-low attachment 96-well round-bottom plates, which allowed for the formation of a single spheroid/well. When the spheroid was formed (48–72 h), Matrigel was added and solidified for 1 h at 37 °C. Images of each spheroid were taken with a microscope (Nikon software NIS Elements) at T0 and T24 h. The Image J software was used to measure the spheres’ area at T0 and the area covered by the invading cells at T24 h. The results represent the ratio T24 h/T0 for each primary culture. When indicated, the spheroids were transfected, with specific siRNA or a scramble control with Lipofectamine RNAi Max (Invitrogen, Waltham, MA, USA). 24 h after transfection, 3D invasion assays were performed as described above.

### 2.3. Western Blot Analysis

Western blots were performed, as previously described [19], using the indicated antibodies: Actin (Millipore, Burlington, MA, USA) and GPM6A (BioLegend, San Diego, CA, USA).

### 2.4. Immunofluorescence Staining and Microscopy

Invasive neurospheres seeded in Lab-Tek chamber slides coated with Matrigel were fixed with 4% PFA for 15 min at RT. Quenching and permeabilization steps were performed using PBS solution containing 5% BSA (Sigma–Aldrich, Burlington, MA, USA) and 0.3% Triton-X100. The primary antibody, anti-GPM6A (Genetex, Irvine, CA, USA), was incubated in PBS 5% BSA and 0.3% Triton-X100 solution for 2 h. The secondary antibody, Donkey anti-Rabbit Alexa Fluor™ 488 (Invitrogen), or Phalloidin–iFluor 594 conjugate (AATBioquest, Sunnyvale, CA, USA) were incubated for 1 h in PBS 5% BSA and 0.3% Triton-X100. Mounting was performed with VECTASHIELD Vibrance^®^ Antifade Mounting Medium with DAPI (Vector Laboratories, Newark, CA, USA). Immunofluorescence stains were analyzed on a Nikon Eclipse Ti with the Nikon software NIS Element AR and on a LSM 880 Fast Airyscan-Zeiss inverted confocal microscope with the Zeiss software Zen 2.

### 2.5. Transfection, RNA Extraction, Reverse Transcription and Real-Time PCR

The scramble control or the specific siRNA against GPM6A, PTPRZ1, or ZEB1 were purchased from Qiagen. Lipofectamine RNAi Max was used for the transfections (Invitrogen). The purification of total RNA was performed with the RNeasy RNA Isolation Kit (Qiagen, Germantown, MD, USA). Reverse transcription was performed using the Prime Script RT Reagent Kit (TAKARA). The ABI-Stepone+ was used for Real-time PCR (Applied Biosystems, Waltham, MA, USA). Normalization was completed with GAPDH.

### 2.6. Genes Correlations

The correlations between GPM6A expression and PTPRZ1 or ZEB1 were performed in Gliovis [20] using the TCGA database and the Pearson correlation coefficients with their associated *p*-values.

### 2.7. 3D Spheroid Formation

GBSC, transfected with specific siRNA or a scramble control, were seeded (500 cells/well) in 96 wells flat bottom plates (6 wells/condition). The number of spheres/well was counted under the microscope after 8–10 days.

### 2.8. 3D Survival Assay under Radiation

GBSC expressing the siRNA (si-GPM6A, si-PTPRZ1, si-Scr) were seeded in 96-well flat-bottom plates (500 cells/wells, 12 wells per condition). Cells were irradiated after 24 h with different doses of X-rays (0 to 6 Gy) using the SmART+ irradiator (Precision X-ray Inc., Madison, WI, USA)., The number of spheres/well were counted 8–10 days post-IR. The calculation of the surviving fraction takes into account the plating efficiency (PE) in the non-irradiated condition (PE = spheres number/seeded cells number × 100).

## 3. Results

### 3.1. Blocking GPM6A Expression Represses Invasion of GBSC

In the first part of this study, we analyzed the invasive capacities of primary cultures of GBSC isolated from 14 patient samples. We performed 3D invasion assays, as described in the Methods. The area covered by the invading cells was measured 24 h after the inclusion in Matrigel. As shown in Figure 1A, we observed high heterogeneity in the invasive capacities of the primary neurospheres derived from the different GB samples. Figure 1B shows some examples of invasion profiles obtained in the 3D invasion assays.

Cells were classified into two groups, highly invasive and low invasive, based on a ratio invading cells area/sphere area greater than 2. Figure 1C shows the significant difference in the invasive capacity between the two groups.

Then, using quantitative PCR, we analyzed the expression of GPM6A and GPM6B in the 14 primary neurospheres and compared the expression levels between the highly and low invasive groups. We observed a significantly higher level of GPM6A in the group of highly invasive GBSC (Figure 2A and Appendix A). On the contrary, GPM6B expression level was not different between the two groups (Figure 2B). Differential expression of GPM6A between highly and low invasive cells was also confirmed at the protein level by Western blot analysis on a panel of highly invasive and low invasive neurospheres (Figure 2C).

We also analyzed the localization of GPM6A in the invasive cells by immunofluorescence. Invasive neurospheres were analyzed 24 h after seeding. We observed a high staining of GPM6A in the neurospheres as well as in the invasive cells (Figure 3A,B). Confocal microscopy analyses showed a punctated staining in lamellipodia/pseudopodia-like structures (Figure 3C,D,E), suggesting a potential role in cell migration.

To our knowledge, the role of GPM6A in GB cells’ invasion has never been published. To investigate if GPM6A could be involved in this process, we performed 3D invasion assays, using invasive primary neurospheres derived from three GB biopsy specimens in which GPM6A was knocked down using a specific siRNA validated for its efficiency to inhibit GPM6A expression in the GB neurospheres compared to a scramble control (Figure 4A,B). As shown in Figure 4C,D, GB spheroids deficient for GPM6A exhibited a significant inhibition of invasion capability relative to control spheroids, confirming the role of this glycoprotein in GBSC invasion.

To assess a potential off-target effect of the GPM6A siRNA, we used a second siRNA and showed in GB neurospheres, a high inhibition of GPM6A expression as well as similar results in 3D invasion assays (Appendix A).

### 3.2. Targeting PTPRZ1 Inhibits GPM6A Expression and GBSC Invasion

We then analyzed, in the glioblastoma database of the Cancer Genome Atlas (TCGA), the correlations between the expression GPM6A and other genes potentially involved in GB cells invasion. One of the strongest correlation observed in the database was with PTPRZ1, an oncogenic protein tyrosine phosphatase highly expressed in GB that has been involved in cell invasion [21]. As shown in Figure 5A, a significant positive correlation was observed, at the mRNA level, between the expression of GPM6A and PTPRZ1 in GB. GPM6A was also positively correlated with the transcription factor ZEB1, which is known for its role in cell invasion, including in GB (Figure 5B). On the contrary, we did not find any positive correlation between GPM6A and the other main transcription factors involved in invasion such as TWIST1, Snail, Slug, or YAP1, whose involvement in the migration of GBSC was previously shown [22,23]. In the GBSC used in this study, we showed that PTPRZ1 and ZEB1 mRNAs, such as those of GPM6A, were preferentially expressed in the group of highly invasive GBSC (Figure 5C,D).

Based on the correlations observed between GPM6A and PTPRZ1 or ZEB1, we tested if the down-regulation of PTPRZ1 or ZEB1 could affect the expression of GPM6A. We observed a decrease in GPM6A mRNA expression in cells transfected with a PTPRZ1 siRNA, which confirmed the direct correlation between the two genes in GB (Figure 5E). In contrast, blocking ZEB1 expression with previously characterized siRNA did not affect GPM6A expression, suggesting a more indirect correlation between ZEB1 and GPM6A (Figure 5F).

In addition, in the primary neurospheres transfected with specific PTPRZ1 siRNAs, validated for their ability to block PTPRZ1 expression, we observed a significant reduction in the invasive capacity of GBSC (Figure 6A).

### 3.3. Down-Regulation of GPM6A or PTPRZ1 Gene Expression Decreases Sphere-Forming Ability of GBSC

Since PTPRZ1 has been previously shown to regulate the proliferation and sphere-forming ability of GB cells [21,24,25], we also analyzed the role of GPM6A in these two processes. The proliferation of GBSC, measured by cell counting, was weakly but significantly decreased after 48 h when the expression of GPM6A was blocked by specific siRNA (Figure 6B). As expected, we also observed an inhibition of GBSC proliferation in cells transfected with PTPRZ1 siRNAs (Figure 6B). The formation of neurospheres was examined in GBSC transfected with GPM6A, PTPRZ1 siRNAs, or a scramble control. Under these conditions, we observed a significant decrease in the number of spheres when GPM6A or PTPRZ1 were blocked with their respective, specific siRNA (Figure 6C,D).

### 3.4. Blocking GPM6A or PTPRZ1 Radiosensitizes GBSC

Radiotherapy is the reference treatment for GB, but the local recurrence, which occurs in almost all cases, highlights the strong radioresistance of GB and GBSC in particular.

To determine whether GPM6A or PTPRZ1 affects radiation sensitivity, we performed 3D survival assays with increasing doses of IR in neurospheres expressing high levels of GPM6A and PTPRZ1. The survival fractions after IR were significantly decreased in GBSC transfected with the specific GPM6A or PTPRZ1 siRNAs compared to the control siRNA, indicating that the down-regulation of their expression radiosensitizes GBSC (Figure 7A,B).

## 4. Discussion

GB is one of the most aggressive brain tumors, being particularly invasive and resistant to radiotherapy. This aggressiveness is essentially due to the presence of tumor stem cells for which there is as yet no targeted clinical therapy [5,6]. Therefore, it is important to understand the mechanisms of invasion and radioresistance of these cells.

In the present study, we demonstrate for the first time the important role of a membrane glycoprotein, GPM6A, in these two processes. Very few data are available on the involvement of GPM6A in cancers. While GPM6A has been identified as a potential oncogene in lymphoid leukemia [12], and contributes to the poor prognosis of colorectal cancer [14,15], its role in GB has never been reported. First, we showed that GPM6A is overexpressed in the invasive GBSC compared to non-invasive cells and localized in lamellipodia/pseudopodia-like structures, suggesting a role in cell migration/invasion. In addition, we demonstrated that blocking its expression in GBSC spheroids with specific siRNA drastically reduces their invasive capacity. Our results are the first to demonstrate the involvement of GPM6A in human tumor cell invasion. Previously, its role in the migration and formation of filopodia had only been demonstrated in primary cultures of neurons expressing endogenous GPM6A, or non-neuronal cells, such as COS-7 or NIH/3T3, transfected with GPM6A [7,8,9,10,11].

The results of our study also show that GPM6A expression is crucial for the formation of spheres by GBSC since the knockdown of GPM6A significantly decreases the number of neurospheres formed by GBSC derived from GB biopsy specimens. These results and those obtained for invasion, support a potential role of GPM6A in the tumorigenicity and aggressiveness of GB.

In a non-tumor context, GPM6A is highly expressed in the CNS and its functions could be dependent on its interaction with laminin [26]. It is interesting to note that GB cells can secrete different forms of laminins, which contribute to tumor progression by playing a role in the invasion and resistance to therapies [27,28,29,30]. Although laminins are known to interact mainly with integrins and mediate their effects, they might also play a role in GPM6A functions in GB.

We also analyzed the potential molecular mechanism regulating the expression of GPM6A in invasive GBSC. In the TCGA database, we showed a very strong correlation between GPM6A expression and a transmembrane tyrosine phosphatase, PTPRZ1. We confirmed the correlation between these two genes in the highly invasive group of GBSC. First, we observed a higher expression level of PTPRZ1 in the invasive cells compared to the non-invasive cells, and secondly, GBSC spheroids deficient for PTPRZ1 exhibited a significantly reduced expression of GPM6A. Similar to GPM6A, PTPRZ1 is highly expressed in the glial cells of the CNS, including astrocytes, oligodendrocyte precursor cells, and oligodendrocytes [31]. PTPRZ1 is also strongly expressed in GB and has been associated with tumorigenicity. This pro-oncogenic phosphatase regulates the proliferation and migration of GB cells and promotes tumor formation. It is also recognized as a stemness marker, which regulates stem-cell-like features and spheres formation [20,23,32,33]. Its high correlation with GPM6A in GB and its involvement in the regulation of GPM6A expression reinforce the hypothesis of a pro-tumoral role of GPM6A in this cancer.

Radiation therapy is one of the standard treatments for GB. However, the intrinsic radioresistance of cancer cells, or the resistance acquired during treatment by adaptation mechanisms, leads to systematic therapeutic failure. Understanding the mechanisms of this resistance could help to identify new therapeutic targets, the inhibition of which could allow radiosensitization.

The role of GPM6A or PTPRZ1 in the radioresistance of GB or other cancer types has never been reported. To our knowledge, this study is the first one demonstrating that GPM6A or PTPRZ1 are involved in the resistance to radiotherapy of GBSC. We showed that targeting GPM6A or PTPRZ1 in GBSC neurospheres, which express high levels of these markers, sensitize cells to radiations.

All of our results suggest that blocking GPM6A or PTPRZ1 could represent an interesting approach in the treatment of glioblastoma, since it would simultaneously target proliferation, invasion, and radioresistance. At present, there is no pharmacological inhibitor capable of blocking GPM6A functions. On the contrary, several laboratories have developed small molecules or monoclonal antibodies targeting PTPRZ1 that are able to suppress GB cell proliferation and migration [24,25,34]. These molecules might also present the possibility to radiosensitize GBSC.

## 5. Conclusions

In summary, our study highlights the overexpression of GPM6A and PTPRZ1 in invasive GBSC and their role in regulating invasiveness, as well as radioresistance. Since there is currently no therapy against GBSC, which are particularly radioresistant and invasive, our study opens perspectives on the potential interest of these biomarkers as therapeutic targets. Their expression may be used to target GBSC invasion, optimize radiotherapy treatments, or to predict the response to radiotherapy.

## Figures and Tables

**Figure 1 cells-11-02128-f001:**
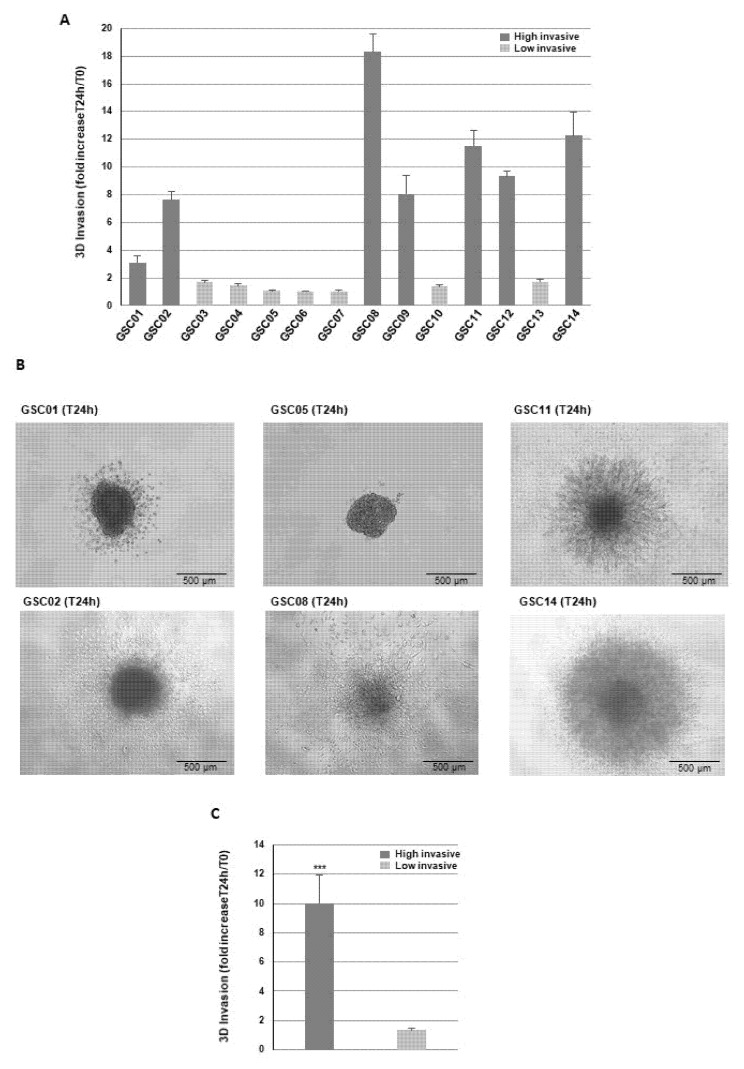
Invasive capacities of GBSC. Primary cultures of GBSC isolated from 14 patient samples were analyzed in 3D invasion assays. (**A**,**C**) Quantification of tumor cells invasion was performed, 24 h after embedding, on 3 independent experiments as described in “Methods”. Cells were classified into two groups, highly invasive and low invasive, based on a ratio invading cells area/sphere area greater than 2. (**B**) Representative micrographs of Matrigel-embedded GB spheroids taken 24 h after invasion into the Matrigel. (**B**) Results are presented as means ± SD. *** *p* < 0.001.

**Figure 2 cells-11-02128-f002:**
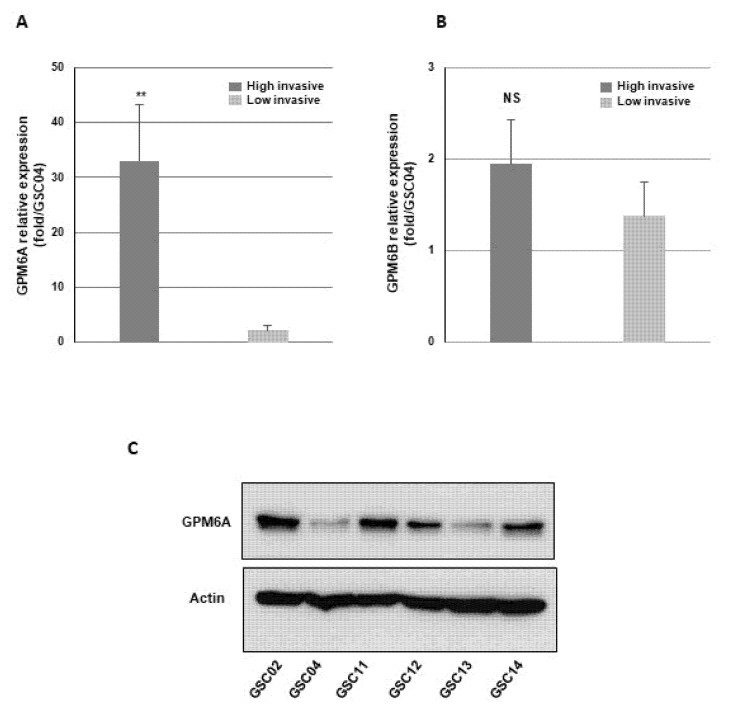
GPM6A is overexpressed in the highly invasive GBSC. mRNA expression of (**A**) GPM6A and (**B**) GPM6B was analyzed by real-time PCR in the 14 primary neurospheres. GAPDH was used for normalization. Results are presented as fold compared to GSC04 expression (which presents the lowest expression). Quantifications between the two groups, highly invasive and low invasive, are presented as means ± SD. ** 0.001 < *p* < 0.01; NS, non-significant. (**C**) GPM6A protein expression was analyzed by Western blot analysis in a panel of highly invasive and low invasive neurospheres.

**Figure 3 cells-11-02128-f003:**
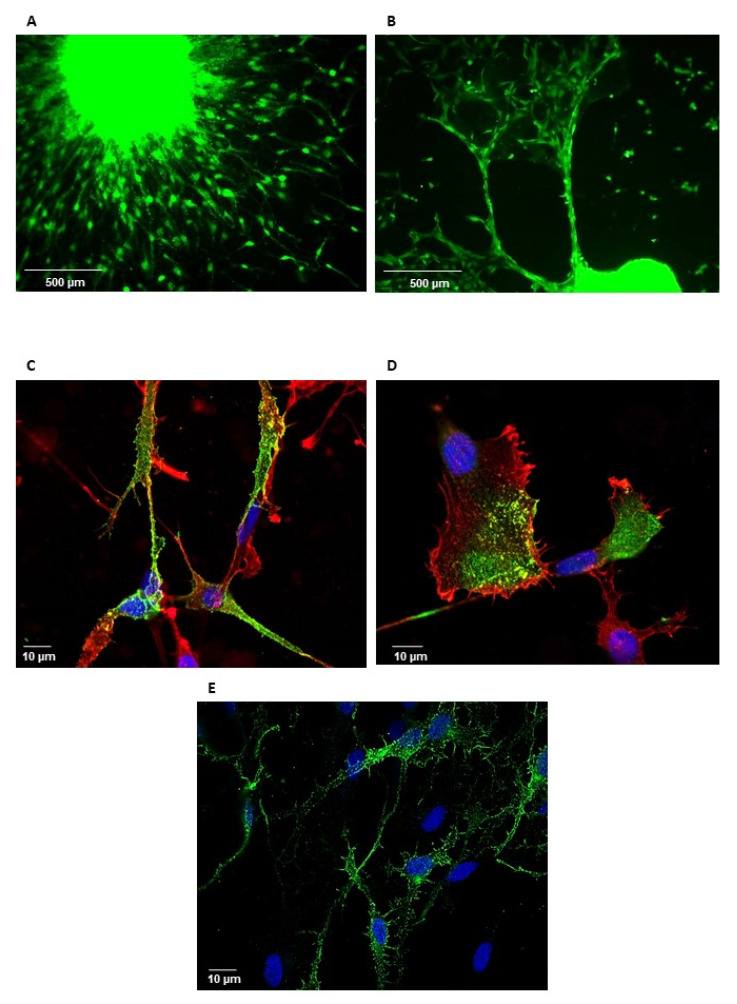
Immunofluorescent staining of GPM6A in invasive GBSC. Invasive neurospheres seeded in Lab-Tek chamber slides coated with Matrigel were immuno-stained with anti-GPM6A antibodies (green) and incubated with phalloidin (red) or Dapi (blue) as described in “Methods”. (**A**,**B**) Immunofluorescence stainings were analyzed on a Nikon Eclipse Ti with the Nikon software NIS Element AR (original magnification 10×). (**C**–**E**) Immunofluorescence stainings were analyzed and on a LSM 880 Fast Airyscan-Zeiss inverted confocal microscope with the Zeiss software Zen 2 (original magnification 60×).

**Figure 4 cells-11-02128-f004:**
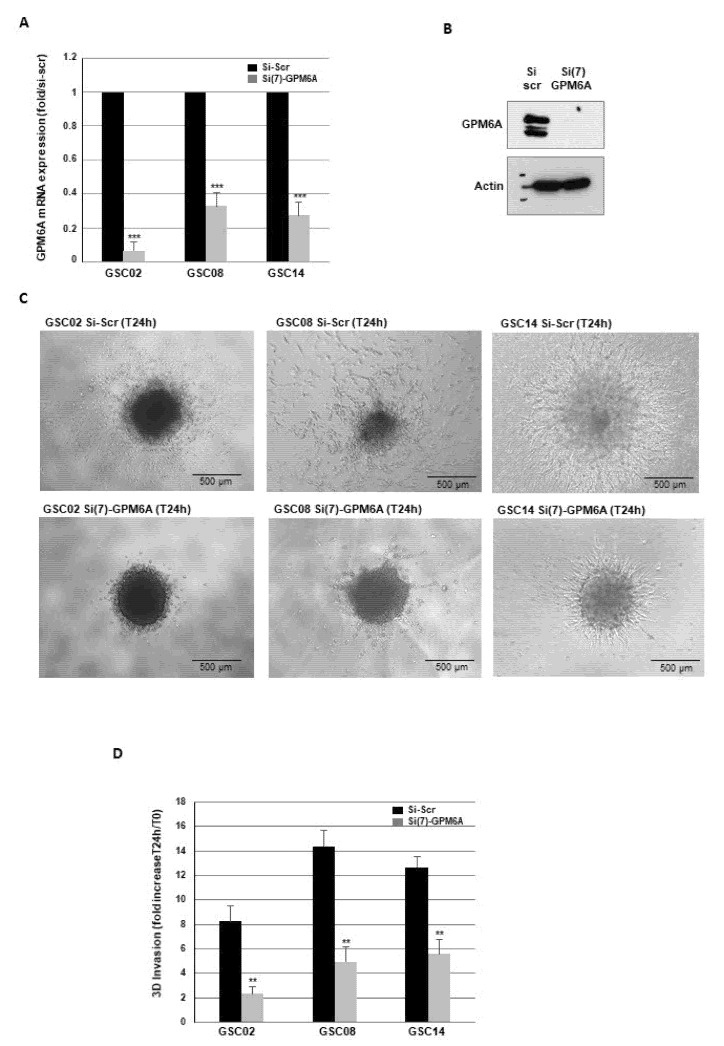
Blocking GPM6A gene expression represses invasion of GBSC. Primary neurospheres from different invasive GB biopsy specimens (GSC02, GSC08, GSC14) were transfected with a specific GPM6A siRNA (si(7)-GPM6A) or a scramble control (si-Scr). GPM6A expression was analyzed by real-time PCR (**A**) or Western blot (**B**). (**C**,**D**) Three-dimensional invasion assays were performed as described in the Methods. Images are representative of three independent experiments. Quantification of 3 experiments are presented as means ± SD. *** *p* < 0.001; ** 0.001 < *p* < 0.01.

**Figure 5 cells-11-02128-f005:**
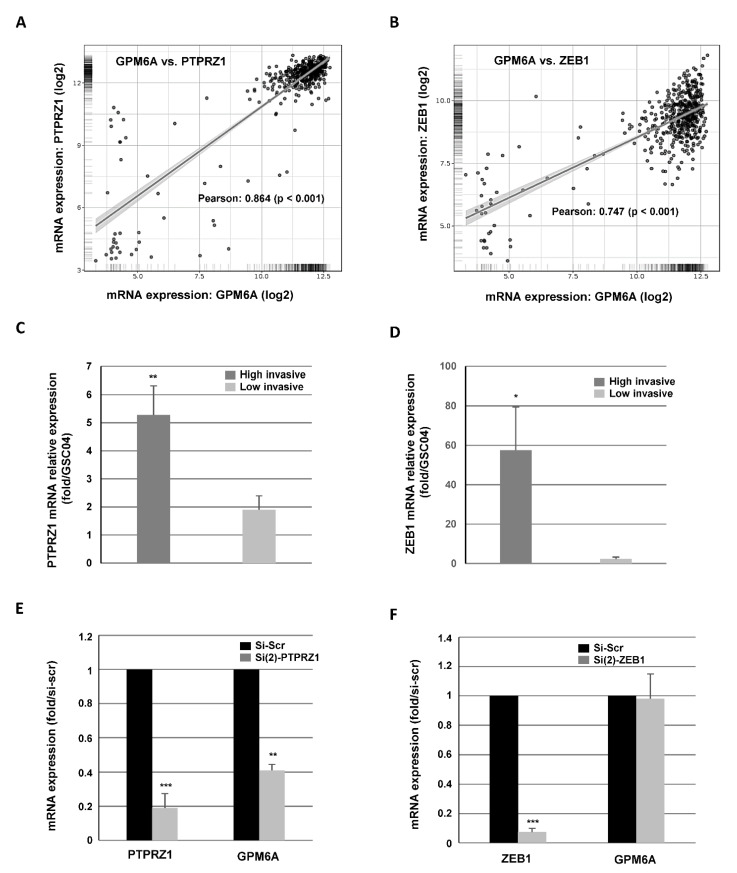
GPM6A and PTPRZ1 expression are correlated in GB. (**A**,**B**) The correlations between GPM6A mRNA expression and PTPRZ1 or ZEB1 were obtained by the co-expression analysis in Gliovis [20] using the TCGA database. Values correspond to the Pearson correlation coefficient and its associated *p*-value. (**C**,**D**) mRNA expression of (**C**) PTPRZ1 and (**D**) ZEB1 was analyzed by real-time PCR in each of the 14 primary cultures of GBSC isolated from 14 patient samples. GAPDH was used for normalization. Results are presented as means of the relative expression (compared to GSC04 which has the lowest expression) for each group (highly or low invasive). Quantifications between the two groups are presented as means ± SD. *** *p* < 0.001; ** 0.001 < *p* < 0.01; * 0.01 < *p* < 0.05. (**E**,**F**) Primary neurospheres were transfected with a specific PTPRZ1 siRNA (si(2)-PTPRZ1), a specific ZEB1 siRNA (si(2)-ZEB1), or a scramble control (si-Scr). PTPRZ1, GPM6A, and ZEB1 mRNA expression was analyzed by real-time PCR. GAPDH was used for normalization. Quantifications of 3 experiments are presented as means ± SD. *** *p* < 0.001; ** 0.001 < *p* < 0.01; * 0.01 < *p* < 0.05.

**Figure 6 cells-11-02128-f006:**
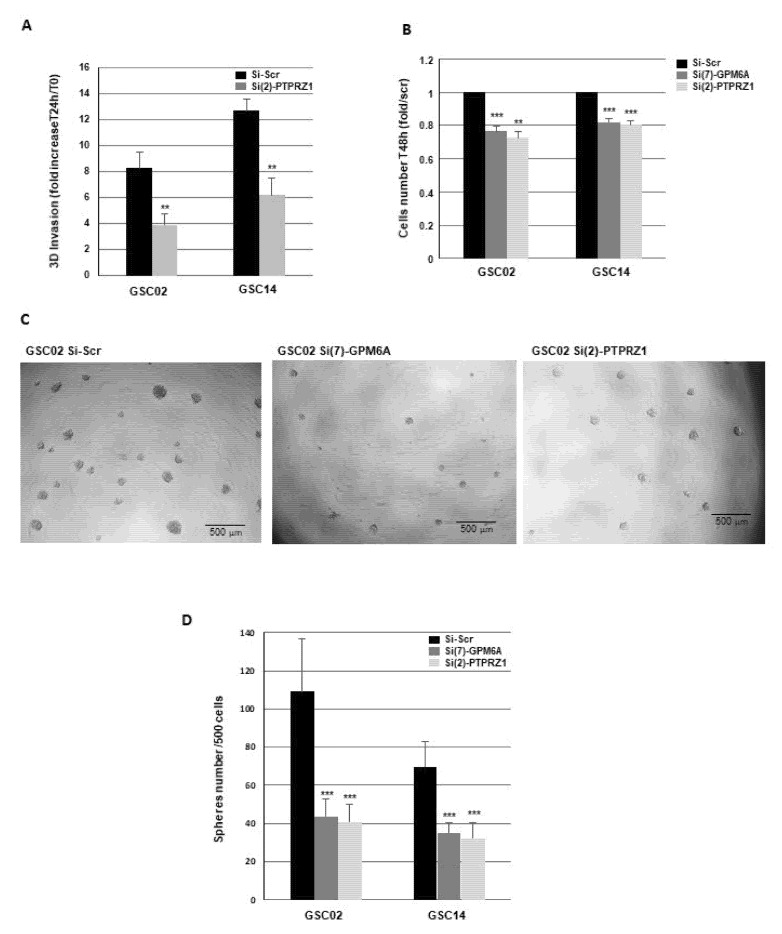
Targeting GPM6A or PTPRZ1 expression decreases invasion and sphere-forming ability of GBSC. (**A**–**D**) GBSC from different GB biopsy specimens (GSC02, GSC14) were transfected with specific PTPRZ1 siRNA (si(2)-PTPRZ1), specific GPM6A siRNA (si(7)-GPM6A), or a scramble control (si-Scr). (**A**) Following transfection, 3D invasion assays were performed as described in the Methods. (**B**) The number of cells was measured by using the cell counter Countess II FL. (**C**,**D**) Following the transfection, cells were seeded in 96-well plates (500 cells/well). After 8–10 days, the number of neurospheres/well was counted under the microscope. (**C**) Micrographs from representative fields were taken (×20). (**A**,**B**,**D**) Quantifications of 3 experiments are presented as means ± SD. *** *p* < 0.001; ** 0.001 < *p* < 0.01.

**Figure 7 cells-11-02128-f007:**
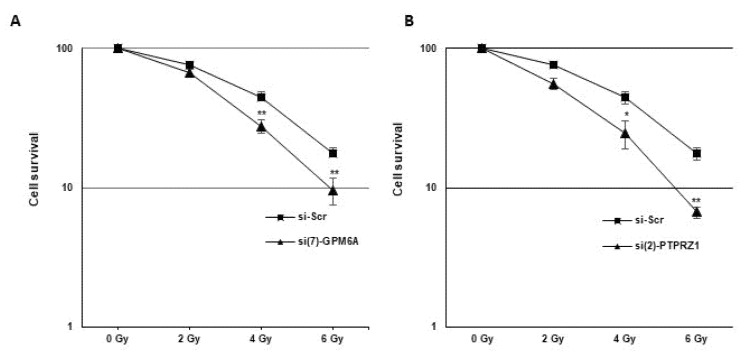
Down-regulation of GPM6A or PTPRZ1 gene expression radiosensitizes GBSC. Primary neurospheres (GSC02) expressing specific siRNAs (**A**) (si(7)-GPM6A or (**B**) si(2)-PTPRZ1) or a scramble control (si-Scr) were used in a 3D survival assays with increasing doses of IR (2 to 6 Gy) as described in the Methods. Quantifications of 3 experiments are presented as means ± SD. ** 0.001 < *p* < 0.01; * 0.01 < *p* < 0.05.

## Data Availability

Not applicable.

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
