# Peer review of "The Glycoprotein M6a Is Associated with Invasiveness and Radioresistance of Glioblastoma Stem Cells"

_cells, 2022, doi:10.3390/cells11142128_

Round 1

Reviewer 1 Report

This is a study evaluating the expression of GPM6A on glioblastoma cancer stem cells. The study is well conducted and the data are well presented as well as the methodology. I think the article could have a greater impact in the literature if some considerations about the glioma clinic were included in the introduction or discussion:

- Recurrence in glioblastoma is an event considered inevitable in pathology however it is not only GSCs that favor tumor regrowth, there are molecular and clinicosurgical variables that favor its return, among these should be mentioned the multifocal forms of GB and the forms with initial periventricular involvement of difficult surgical access rather than genetic background, (it is possible to mention, Armocida D, Pesce A, Di Giammarco F, Frati A, Salvati M, Santoro A. Histological, molecular, clinical and outcomes characteristics of Multiple Lesion Glioblastoma. A retrospective monocentric study and review of literature. Neurocirugia (Astur : Engl Ed). 2021 May-Jun;32(3):114-123. English, Spanish. doi: 10.1016/j.neucir.2020.04.003. Epub 2020 Jun 18. PMID: 32564972. AND Armocida D., Pesce A., Palmieri M., et. al. Periventricular zone involvement as a predictor of survival in glioblastoma patients: A single-center cohort-comparison investigation concerning a distinct clinical entity, Interdisciplinary Neurosurgery, Volume 25, 2021, 101185, ISSN 2214-7519, https://doi.org/10.1016/j.inat.2021.101185.)

- I would insert a concluding note;

- In view of the new WHO classification, it is suggested to no longer name Glioblastoma Multiforme (GBM), but only Glioblastoma (GB);

Reviewer 2 Report

Comment:

After careful consideration, I feel that it has merit but is not suitable for publication as it currently stands (only in vitro). I uphold the standard and integrity of [Cells] with an impact factor of ~ 6.4, which is substantially higher than the average Journals of IF 1.0 in SCI WOS databases [29,000 Journals]. Some 19 issues came to my attention, as follows.

Specific comments:

1)    Lines 73-77: “The GBMSC used in the study mesenchymal: GSC08, GSC10, GSC14 and 11 proneural: GSC01, GSC02, GSC03, GSC04, 74 GSC05, GSC06, GSC07, GSC09, GSC11, GSC12, GSC13) have been validated for their ability of self-renewal, the overexpression of stem cell markers, their pluripotent aptitude to differentiate into neural lineages and their high tumorigenic potential in vivo.”

Any karyotype data? What was their stem cell biomarker data set?

2)    Line 79: typo. “2.23. D tumor spheroid Invasion assay.”

3)    Lines 79-88: What was the control cell line? Why did they not use the clonogenic assay (doi: 10.1186/1475-2867-12-41)? Lines 130-137: 2.9 3D clonogenic assay. Could they combine and compare as these two matrices are different in their tissue elasticity (doi: 10.1371/journal.pone.0120336)?

4)    Lines 140-141: “from 14 human GBM biopsy specimens cultured as primary neurospheres” – could you table all 14 specimens for the characterization?

5)    Fig 1. How did they pick up how many spheroids? How did they know the cell density (thickness, diameter)? E.g., GSC01(plus GSC5) is much thicker than GSC08 (plus GSC14). Why? Did they measure only the horizontal dimension, not the vertical dimension? Vertical growth was also invasive. Fig 1C: How did they draw the line to define high vs. low invasiveness? Any alternative measurement?

6)    Fig 2. Line 168: Results are presented as fold compared to GSCO4 expression (low) – which specimen was for high? How did both mRNA and protein coordinate in quantification?

7)    Fig 3. Scale bars for all panels are needed. What was the fluorophore for anti-GPM6A?

8)    Line 186: “The role of GPM6A in GBM cells invasion has never been studied.” On what reasoning did they do these?

9)    Fig 4C is equal to Fig 1B for CSC14. Why?

10)  Any studies of FACS on those neurosphere compositions? What percentage of those biomarkers of different specimens?

11)  Lines 205-218: glioblastoma database of the Cancer Genome Atlas, (TCGA). Did they realize the difference between bulk tumors and cultured neurosphere cells? How did they deal with cultured artifacts? How could they tear down the heterogeneity of the bulk tumor (doi: 10.1093/carcin/bgy052.)?

12)  Figure 5. GPM6A and PTPRZ1 expression are correlated in GBM. How did these two relate? Protein expression?

13)  Line 224: analyzed by real-time PCR in the 14 primary neurospheres? What did they mean with such panels? All of 14 or a specific one?

14)  Fig6D: Cell numbers of neurospheres? How many neurospheres did they pick up by what standard?

15)  Fig 6. Line 246: 3D invasion assays were performed as described in « Methods ». Note that this is not qualified as invasive assays; neither control nor quantification is justified.

16)  Line 263: both proteins in the sphere-forming ability of GBMSC (Fig. 6C, 6D). Where did they show protein data? Not in Fig 6.

17)  Discussion: blocking GPM6A or PTPRZ1 could represent” should be illustrated with a schematic diagram.

18)  Lines 344-346: “GPM6A and PTPRZ1, two proteins whose expression is correlated in GBM databases, are overexpressed in invasive GBMSC compared to non-invasive cells and contribute to GBMSC invasion and radioresistance.” Where is PTPRZ1 protein? How did they correlate in GBM databases at the protein level? Data?

19)  Most importantly, did they back-track the patient data (pathology, MRI) to see if the in vitro data relate (relevance)? 

Round 2

Reviewer 2 Report

Revision 1 has incorporated all comments, so I accepted it for publication.